# Multifractal Analysis of River Networks under the Background of Urbanization in the Yellow River Basin, China

**Jinxin Wang** [1], **Zilong Qin** [2,*], **Yan Shi** [2] and **Jing Yao** [2]

1   School of Earth Science and Technology, Zhengzhou University, Zhengzhou 450001, China; jxwang@zzu.edu.cn
2   School of Water Conservancy Engineering, Zhengzhou University, Zhengzhou 450001, China; shiyan2019@gs.zzu.edu.cn (Y.S.); yaojing@gs.zzu.edu.cn (J.Y.)
*   Correspondence: qinzilong@gs.zzu.edu.cn; Tel.: +86-188-3912-6892

**Abstract:** Multifractal theory provides an effective method for the scientific quantification of the river network features. This method has been applied to estimate river network structure in previous research, but there are few temporal and spatial analysis studied for large river basins based on multifractals. In this study, the Yellow River Basin and its nine provinces were selected as the study area, and the generalized fractal dimension and multifractal spectrum of the study area during the period 2000–2020 were calculated and analyzed. We analyzed the response relationship between the river network changes and the urbanization process. Results indicated that the river network of the study area exhibited obvious multifractal properties, which were mainly affected by dense river networks. The complexity of the studied river network has shown a tendency to decrease over time. The rate of change in the river network structure has a high positive correlation with urbanization, that is, the higher the rate of urbanization, the greater the impact on the river network structure. Additionally, the changes of the river network structure are more affected by urbanization during the rapid urbanization stage. We applied multifractal analysis to study the river network structure changes, which is of great significance for scientifically quantifying fluvial characteristics and studying the development and evolution of river networks.

**Keywords:** multifractal analysis; river network; quantitative description; urbanization; Yellow River Basin

## 1. Introduction

The structure of a river network is one of the most basic components of basin geomorphology, and it is affected by many factors. Accurate and quantitative expression of river network structures is very important for studying river network sedimentation processes [1], extreme hydrological events in river basins [2], and river network development [3]. Although some traditional statistical methods can reveal some characteristics of river network structure, such as river density [4,5], branch ratio [6], and total length, owing to the high complexity of a river network structure, these methods usually cannot well describe certain key features of river networks [7]. The fractal theory provides an effective means of quantitatively describing the characteristics of river networks.

Fractal theory was first proposed by Mandelbrot to measure the length of the British coastline [8]. Fractal theory includes simple fractals and multifractals. Although the simple fractal can describe the complexity of a research object to a certain extent, simple fractal only uses a fractal dimension value to generalize the characteristics of a region, and it cannot reflect the local features [9,10]. Subsequently, with the further development and improvement of the multifractal theory, this problem has been resolved [11]. The multifractal theory has been widely applied in many fields since it was proposed. River basin networks have intricate and obvious structural properties, including self-similarity and fractal characteristics [12]. Multifractal theory has been successfully applied to study

the characteristics of river networks [13,14]. River network structural information is often extracted from the digital elevation model (DEM). Combined with multifractal analysis, the relationship between the flow accumulation threshold and the resolution of the DEM when extracting river network structural information has been effectively studied. In other words, the flow accumulation threshold increased with an increase in DEM resolution [15]. Zhang et al. [16] improved the results of extracting river networks from DEM in combination with multifractals and concluded that the multifractal method has a better effect in determining the flow accumulation threshold than the simple fractal method of box counting. As one of the indicators of the complexity of a river network structure, the multifractal spectrum has not only been well applied in the study of river network structure and changes but has also been used to improve the predictability of runoff and rainfall models [17]. Moreover, it has been used to study the control effect of lithology on the multifractal characteristics of river networks [18]. At present, the methods used to calculate the multifractal characteristics of river networks mainly include the fixed-size algorithm (FSA) and the fixed-mass algorithm (FMA). However, it has been found that the fixed-size algorithm is more suitable for calculating the multifractal characteristics of a river network [19], and it is widely applied in multifractal analysis.

The development of a river network is a result of natural processes, which are affected by many factors such as precipitation, temperature, topography, and geology. However, in recent decades, the structure or development of many river networks has been greatly affected by large-scale urban expansion, which has had a strong anthropogenic impact on the surface of the earth [20]. Under the influence of urbanization, more than half of the world's river network structures have undergone changes to varying degrees [21]. Rapid urbanization has led to adverse effects such as rapid changes in river network structure and the frequent occurrence of extreme hydrological events such as flood disasters [22–24]. While the river network structure has been the basis for further research on the topographical and hydrological characteristics of river basins [25], studies should also include the changes in river networks in the context of urbanization. In addition, there are few applications of the multifractal theory on large basins of quantitative research, and multifractals can more comprehensively quantitatively reflect the local characteristics of the river network. Therefore, it is of great significance to study the relationship between the urbanization of large river basins and river network based on the multifractal method and the influence of urbanization on river network structure.

Multifractal analysis quantitatively expresses the features of river networks by calculating generalized the fractal dimension and the multifractal spectrum, which can better reflect the local characteristics of river networks in large river basins. In this study, the Yellow River Basin was selected as the study area, as the Yellow River is the second longest river in China and the goal of high-quality development of the Yellow River basin has been put forward in recent years. It is of great scientific value to study the temporal and spatial variation characteristics of the river network structure. The purpose of this paper is to (1) test the applicability of multifractal theory in the quantitative analysis of river network characteristics in the large basin, (2) study the multifractal characteristics of the river network in the Yellow River Basin over the past 20 years, and (3) analyze the temporal and spatial variations of the river network structure in the study area, (4) finally revealing the relationship between the urbanization process and changes in the river network structure.

## 2. Materials and Methods

### 2.1. Study Area

The study area is located between 32° N, 96° E and 42° N, 119° E, and contains more than 370 counties in nine provinces of China (Figure 1): Qinghai Province (QH), Sichuan Province (SC), Gansu Province (GS), Ningxia Hui Autonomous Region (NX), Inner Mongolia Autonomous Region (NMG), Shaanxi Province (SA), Shanxi Province (SX), Henan Province (HN), and Shandong Province (SD). The Yellow River begins in the Bayan Har Mountains in the Qinghai Province of China and empties into the Bohai Sea. It is

5464 km long with a basin area of 795,000 km² and constitutes the fifth longest river in the world and the second longest in China [26]. The Yellow River basin encompasses a vast area with many mountains and has a great difference in elevation from east to west (about 4800 m). The landforms of the various regions associated with the river basin are also very different. Furthermore, the basin is in the middle latitudes, and so it is more affected by atmospheric changes and monsoon circulation. Thus, there are significant differences in climate within the basin. The temperature difference across the Yellow River Basin is very large. In general, as the terrain of the three-tiered elevation changes from west to east, the temperature changes from colder to warmer. The east–west temperature gradient is notably greater than the north–south gradient. The annual temperature difference in the basin is also relatively large. The general trend is that the annual average temperature in the area north of 37° N is between 31 °C and 37 °C, while the temperature in the area south of 37° N is mostly between 21 °C and 31 °C. Precipitation in the Yellow River Basin is concentrated, unevenly distributed, and has significant inter-annual variation. The annual precipitation in most of the regions is between 200 and 650 mm. It is greater than 650 mm south of the middle of the upper and lower reaches, while the precipitation in the deep inland areas of the Ningxia Hui and Inner Mongolia Autonomous Region in northwestern China is less than 150 mm. The basin has a strong evaporative capacity, with an average annual evaporation of 1100 mm. The upper reaches of Gansu Province, Ningxia Hui Autonomous Region, and the central and western regions of the Inner Mongolia Autonomous Region have the largest annual evaporation in China. The maximum annual evaporation in these regions can exceed 2500 mm.

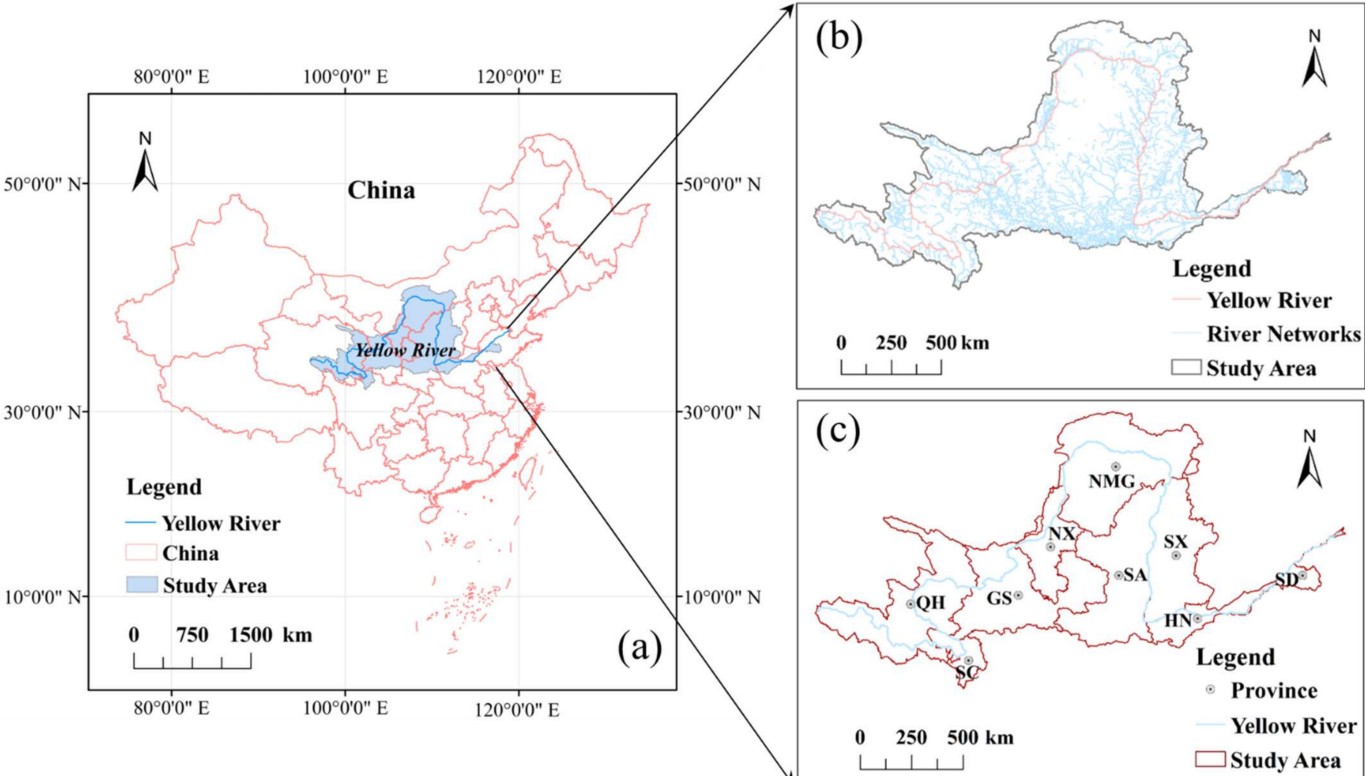

**Figure 1.** Study Area: (**a**) Location of the Yellow River Basin in China; (**b**) the river network of the Yellow River Basin; (**c**) The provinces included in the Yellow River Basin: Qinghai Province (QH), Sichuan Province (SC), Gansu Province (GS), Ningxia Hui Autonomous Region (NX), Inner Mongolia Autonomous Region (NMG), Shaanxi Province (SA), Shanxi Province (SX), Henan Province (HN), and Shandong Province (SD).

## 2.2. Data Description

The first step in calculating the multifractal features of the study area during different periods was the extraction of the river networks. Digital data of the studied river networks during the 2000s and the 2010s were obtained by digitizing 1:50,000 digital line graphics (DLGs). For the 2020s, data were derived from OpenStreetMap (OSM, https://www.openstreetmap.org/ (accessed on 20 August 2021)). We first identified the river data from the digital line graphics and then extracted them using ArcGIS 10.4. To ensure accuracy of the river network vector data, the river network of the study area was corrected by adding and deleting the river networks in three periods based on Google Earth remote sensing images. Based on the above data and the boundary vector data of the Yellow River Basin provided by the National Cryosphere Desert Data Center (http://www.ncdc.ac.cn (accessed on 20 August 2021)), the river networks of the Yellow River Basin in three periods were clipped using ArcGIS 10.4.

## 2.3. Method

In this study, the multifractal method was used to analyze the spatiotemporal variation characteristics of the Yellow River Basin during different periods. Multifractal theory has been widely used since it was first proposed. Compared to mono-fractals, multifractals can describe characteristics of the river networks at different scales. Therefore, multifractals can be defined as the collection of a series of mono-fractals, each having its own singularity exponent $\alpha$ (Lipschitz–Hölder or scaling exponent) and mono-fractal dimensions [27,28]. Multifractal dimensions ($D_q$) and the multifractal spectrum ($f(\alpha)$) were used to describe the features of the study area. The spectrum yielded the dimensions of the fractals with the same singularity exponent [29]. We adopted the fixed-size box-counting method [17] to calculate the multifractal characteristics of the study area. The study's multifractal analysis flowchart and urbanization analysis method are shown in Figure 2.

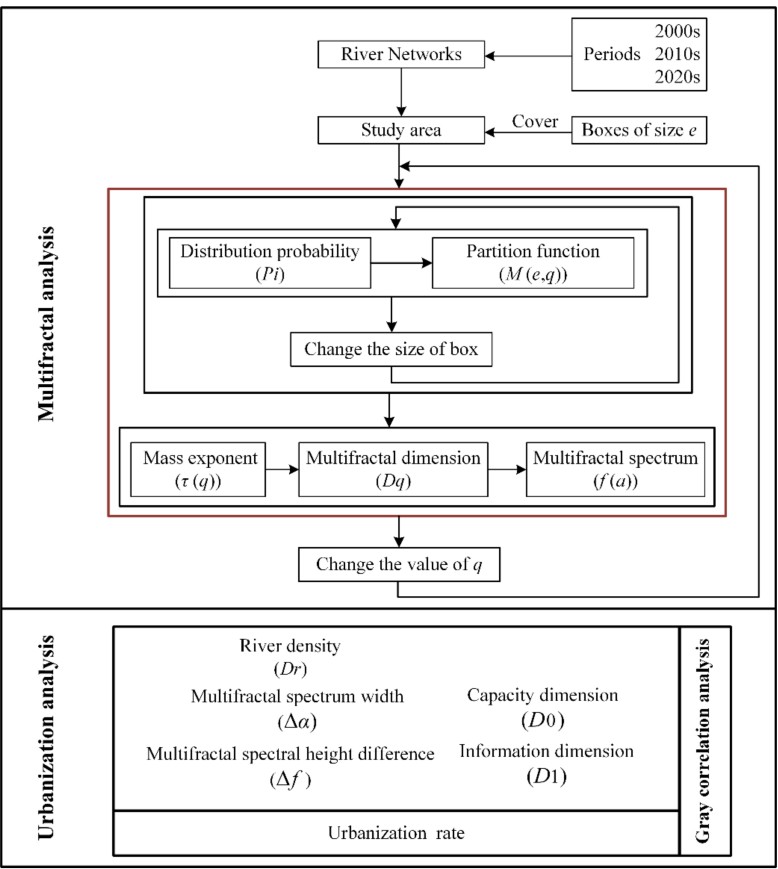

**Figure 2.** The study methodologies.

The steps for calculating the multifractal measures of the study areas are as follows:

(1)　The study area $F$ is covered with boxes of size $e \times e$, and the total number of non-empty boxes is denoted $N(e)$. $p_i(e)$ is the probability measure of the region contained in each box; that is, the distribution probability of the characteristic information. $p_i(e)$ differs for different units. $p_i(e)$ and $e$ are related via Equation (1):

$$p_i \propto e^{\alpha}, \tag{1}$$

where $\alpha$ is a singular exponent that corresponds to different units and is determined by the probability measure ($p_i(e)$). $p_i(e)$ can be computed using Equation (2):

$$p_i(e) = \frac{c_i}{\sum_{i=1}^{N(e)} c_i}, \tag{2}$$

where $c_i$ is the characteristic information of the river network in box $i$ (e.g., river length), and $\sum_{i=1}^{N(e)} c_i$ is the total characteristic information of the study area under the scale of $e$, which reflects the overall morphological features of the studied basin. In this study, $c_i$ is the sum of the river lengths in box $i$, and $\sum_{i=1}^{N(e)} c_i$ is the total length of the study area.

(2)　The partition function $M(e,q)$ is defined as the weighted sum of the slope distribution probability $p_i(e)$ to power $q$ (Equation (3)):

$$M(e,q) = \sum_{i=1}^{N(e)} p_i^q(e), \tag{3}$$

where $q$ is the order of the statistical moment, $q \in (-\infty, +\infty)$, and is used to describe the magnitude of the singularity in the multifractal analysis, and a different $q$ represents the important role played by different river network probability subsets in the partition function. In the calculation, we took different $q$ values and calculated the partition function $M(e,q)$ under the corresponding $q$ value. There is a good linear relationship between the logarithm of the partition function $\ln M(e,q)$ and the logarithm of the box size $\ln e$ when the networks are multifractal in nature. This is an important basis for judging whether a research object has multifractal properties.

(3)　For a given moment $q$, the relationship between the mass exponential function $\tau(q)$ and $M(e,q)$ is given by Equation (4). In the calculation, the size of the box $e$ under the corresponding $q$ value is changed, and the partition function under the corresponding box size is computed. Then, $\tau(q)$ can be computed through the coefficient of the straight line fit of $\ln M(e,q) \sim \ln e$ (Equation (5)). With the change in $q$, the corresponding $\tau(q)$ can be calculated using the above procedure.

$$M(e,q) \propto e^{\tau(q)}, \tag{4}$$

$$\tau(q) = \lim_{e \to 0} \frac{\ln M(e,q)}{\ln e}, \tag{5}$$

where $\tau(q)$ is the eigenvalue of the multifractal behavior. When $\tau(q)$ is a convex function with respect to $q$, the research object exhibits multifractal features. This is another important criterion for judging whether a research object has a multifractal property.

(4) The generalized fractal dimension $D_q$ is defined by Equation (6) and varies with $q$. $D_q$ can reflect the singularity of each subset of the research object from an overall perspective, so there is the relationship between $D_q$ and $\alpha$ in Equations (7) and (8).

$$D_q = \begin{cases} \frac{1}{q-1}\lim\limits_{e\to 0}\frac{\ln M(e,q)}{\ln e} = \frac{\tau(q)}{q-1} & q \neq 1 \\ \lim\limits_{e\to 0}\frac{\sum_{i=1}^{N(e)} p_i \ln p_i}{\ln e} & q = 1 \end{cases}, \tag{6}$$

$$\lim_{q\to+\infty} D_q = \alpha_{\min}, \tag{7}$$

$$\lim_{q\to-\infty} D_q = \alpha_{\max}, \tag{8}$$

Here, $D_q$ is usually a monotonically decreasing function with $q$, and the typical generalized fractal dimension curve is shown in Figure 3. When $q = 0$, $D_{q=0}$ represents the capacity dimension in the multifractals. When $q = 1$, $D_{q=1}$ represents the information dimension. When $q = 2$, $D_{q=2}$ represents the correlation dimension. $D_q$ describes the scaling behavior of the region where the probability measures are most concentrated when $q \to +\infty$ and most rarefied when $q \to -\infty$.

(5) When $\tau(q)$ is differentiable, the multifractal spectrum $f(\alpha)$ and singular exponent $\alpha(q)$ can be obtained by the Legendre transformation of Equation (9).

$$\begin{cases} \alpha(q) = \frac{d\tau(q)}{dq} \\ f(\alpha) = q\cdot\alpha(q) - \tau(q) \end{cases}, \tag{9}$$

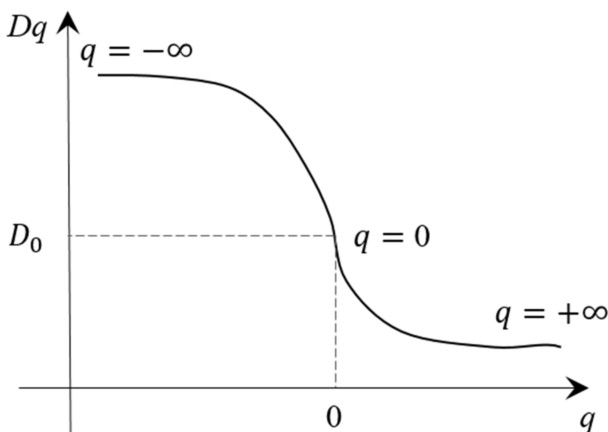

**Figure 3.** The typical generalized fractal dimension diagram.

Here, $f(\alpha)$ is usually a smooth, upward convex curve. Each point on the $f(\alpha){\sim}\alpha(q)$ curve represents the fractal dimension of the subset with the same singular exponent $\alpha(q)$ [30,31]. The $f(\alpha){\sim}\alpha(q)$ curve is converted to a point in the mono-fractal.

Three parameters in the multifractal spectrum are important when describing the heterogeneity of river networks (Figure 4):

- The span of the singular exponent $\alpha(q)$ is the width of the multifractal spectrum, $\Delta\alpha$ (Equation (10)). $\alpha(q)$ indicates the degree of fluvial inhomogeneity, irregularity, and complexity in each sub-region within the basin. $\alpha_{\min}$ and $\alpha_{\max}$ (Equations (7) and (8)), respectively, indicate the singular exponent of the distribution probability of the maximum characteristic information $p_i(e)_{\max}$ and the distribution probability of the minimum characteristic information $p_i(e)_{\min}$ with the change in $e$. The smaller the $\alpha_{\min}$, the larger is the $p_i(e)_{\max}$. Therefore, we can use the span of the singular exponent $\Delta\alpha$ to describe the unevenness in the distribution probability of the river network. A larger $\Delta\alpha$ indicates that the distribution of characteristic information in the basin is less uniform, the internal difference in the research object is greater, and

the polarization trend of each subset probability is clearer. In contrast, a smaller $\Delta\alpha$ indicates that the difference is smaller inside the fractal body, and the distribution of subsets tends to be concentrated and uniform.

- The difference between the maximum and minimum values of the multifractal spectrum is $\Delta f$ (Equation (11)). $f(\alpha_{\min})$ and $f(\alpha_{\max})$ represent the number of subsets of the maximum and minimum probabilistic characteristic information, respectively. The difference in $\Delta f$ can be used to calculate the difference between the maximum and minimum distribution probability subset numbers of the basin characteristic information. When $\Delta f < 0$, the curve $f(\alpha)\sim\alpha(q)$ is hooked to the right, and the number of grid points contained in the maximum characteristic information distribution probability subset is less than the minimum probability subset number. The river network is densely distributed. In contrast, when $\Delta f > 0$, the curve is hooked to the left. When $\Delta f = 0$, the curve $f(\alpha)\sim\alpha(q)$ is symmetrical and bell-shaped.
- Symmetry of curve $f(\alpha)\sim\alpha(q)$. The multifractal spectrum is more symmetrical, which indicates that the fluvial distribution proportion is more uniform in the study area.

$$\Delta\alpha = \alpha_{\max} - \alpha_{\min}, \tag{10}$$

$$\Delta f = f(\alpha_{\min} - \alpha_{\max}), \tag{11}$$

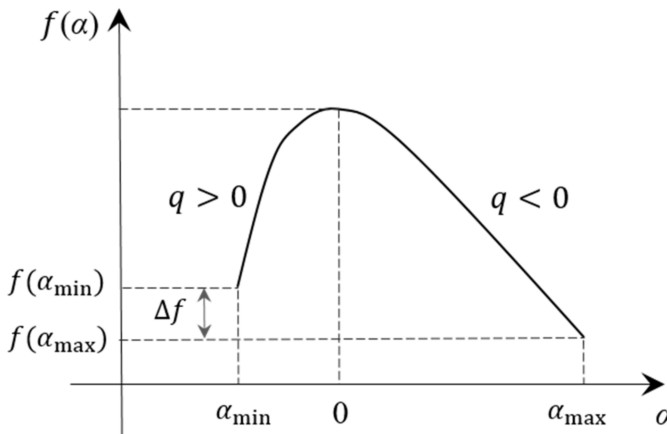

**Figure 4.** The typical multifractal spectrum diagram.

(6) When calculating the generalized fractal dimension $D_q$ and the multifractal spectrum $f(\alpha)$, the value of $q$ plays an important role in the accuracy of the calculation results [32–34]. Theoretically, $q \in (-\infty, +\infty)$, but in the actual calculation, only a limited range can be selected as the value of $q$. According to the research of [35], when the convergence coefficient $\zeta < 0.2\%$, the resulting changes to $\frac{d\alpha_{\max}}{\Delta\alpha}$ and $\frac{d\alpha_{\min}}{\Delta\alpha}$ are very small. The multifractal spectrum calculated within this range can be considered as a multifractal spectrum that reflects the characteristics of the research object. The value range of $|q|$ can be calculated using (12):

$$\zeta = \frac{|f_q - f_{q-1}|}{|f_q - f(\alpha)_{\max}|}, \tag{12}$$

In this study, the range of $q$ was set to [–100, 100], and the step was $\Delta q = 1$. Then, according to Equation (12), the river networks of the Yellow River Basin in the 2000s, 2010s, and the 2020s were calculated to determine the value range of $q$. The range of $q$ was calculated as follows: When the convergence coefficient $\zeta < 0.2\%$, the value of $q$ in both the 2000s and the 2010s is $[-21, 21]$, the value of $q$ in the 2020s is $[-18, 18]$, and the step is $\Delta q = 1$. When the value of $q$ is beyond the calculated range, the results no longer satisfy the geometric characteristics of the multifractals.

## 3. Results

### 3.1. Determination of Multifractal Characteristics

Before conducting a multifractal analysis, it is necessary to assess whether the studied river network has multifractal properties. In this study, the logarithmic curve of the partition function $M(e,q)$ and the box size $e$ in the Yellow River Basin were calculated for the 2000s, 2010s, and the 2020s. The scale-free interval of the study area was also calculated. That is, the range of the box size was 500 to 40,000 m with an increment of 500 m. To show the calculation results clearly, we exhibit only five values for the Yellow River Basin in each of the three periods. For the 2000s and the 2010s, the results are exhibited for $q$ values of $-21$, $-10$, $0$, $10$, and $21$. For the 2020s, the results are exhibited for $q$ values of $-18$, $-9$, $0$, $9$, and $18$ (Figure 5). As shown in Figure 5, when $q < 0$, the curve cluster fluctuates slightly. In contrast, when $q \geq 0$, the curve is more stable, and closer to the fitting straight line. Overall, the $\ln e$ and $\ln M(e,q)$ of the river networks in the Yellow River Basin during the three periods have a good linear relationship, which satisfies the exponential relationship of Equation (4). The linear correlation and determination coefficient ($R^2$) of the curve cluster were greater than 0.96. These results indicate that the study area has scale invariance within the selected scale range. That is, the river networks of the Yellow River Basin in the three periods have obvious multifractal properties.

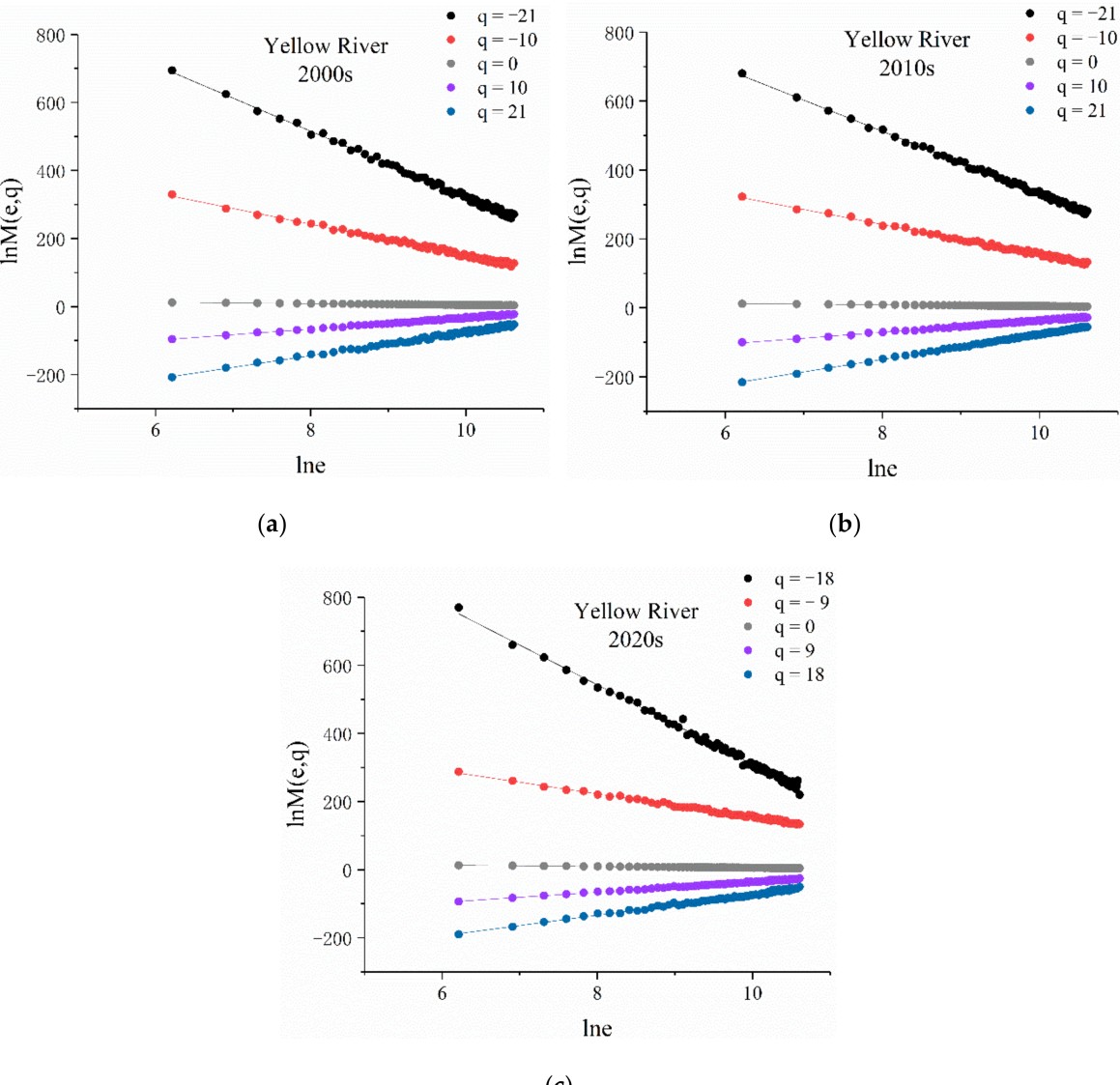

**Figure 5.** The relationship between $\ln e$ and $\ln M(e,q)$ of the Yellow River Basin in the (**a**) 2000s, (**b**) 2010s, and (**c**) 2020s.

Another basis for judging whether the river network has multifractal properties is whether $\tau(q)$ is a convex function with respect to $q$. The relationship between the order moment $q$ and the mass exponent $\tau(q)$ of the Yellow River Basin for the three periods is shown in Figure 6. The trend lines are all upward convex curves in the studied periods, and $\tau(q)$ increases with an increase in $q$. When $q > 0$, the rate of increase in the mass exponent $\tau(q)$ decreases. The mass exponent of the river network in the study area during the three periods increased with time ($(\tau(q))_{2020} > (\tau(q))_{2010} > \tau(q)_{2000}$). The change in the 2010s was less than that in the 2000s, but the mass exponent curves of the two periods were similar. The change in the 2020s was greater than that of the previous two periods. The convex curve in Figure 6 also proves that the river network of the Yellow River basin has a multifractal nature in each of the three periods.

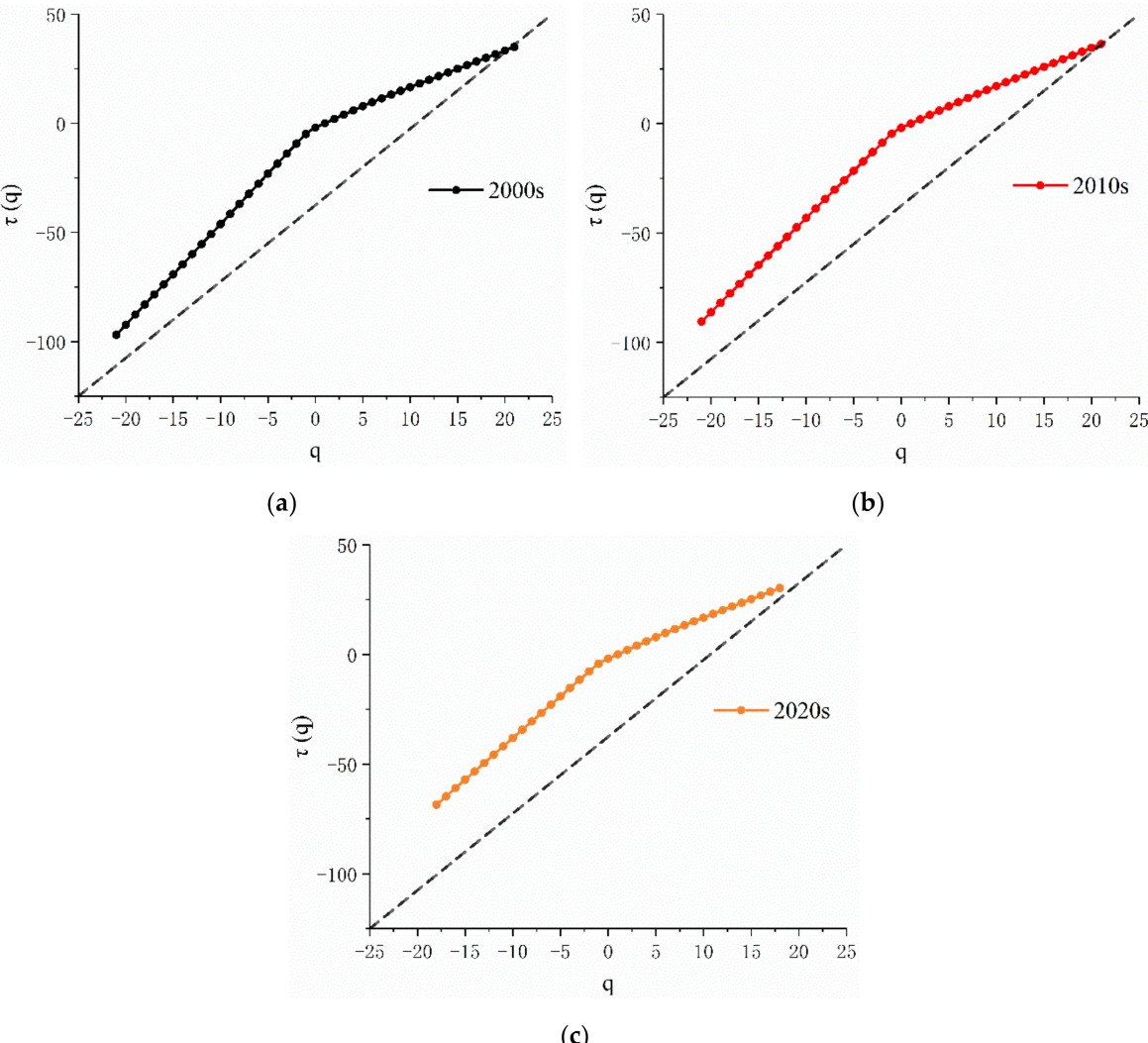

**Figure 6.** The relationship between $\tau(q)$ and $q$ of the Yellow River Basin in the (**a**) 2000s, (**b**) 2010s, and (**c**) 2020s.

### 3.2. Multifractal Dimension Analysis

The spectrum of the generalized multifractal dimension, $D_q$ was computed using the least square linear regression. For the 2000s, we calculated the generalized multifractal dimension of the Yellow River Basin for $q$ values of $-21$ to 21 (step length $\Delta q$ is 1). For the 2020s, calculations were performed for $q$ values of $-18$ to 18 (step length $\Delta q$ is 1). The calculation results are presented in Figure 7. When $q > 1$, the generalized multifractal dimension $D_q$ describes the properties of regions with higher or more concentrated probability measures. When $q < 1$, $D_q$ describes the properties of the regions with lower or more

sparse probability measures. As shown in Figure 7, $D_q$ decreases as the order moment $q$ increases. When $q < 1$, $D_{q2000} > D_{q2010} > D_{q2020}$, which indicates that the complexity of the river network decreased over time in the study area with a low density of the river network. When $q > 1$, the $D_q$ value of the Yellow River Basin is the smallest. The curve drops most rapidly in the 2020s from a maximum value in the 2010s. The change is relatively stable, indicating that the complexity of the river network is also reduced to a certain extent in the areas with high river network density. When $0 < q < 3$, the multifractal dimension curves of the three periods coincide. These results reflect the decreases in the number and density of the river network, as well as the simplification of the river network over the last 20 years.

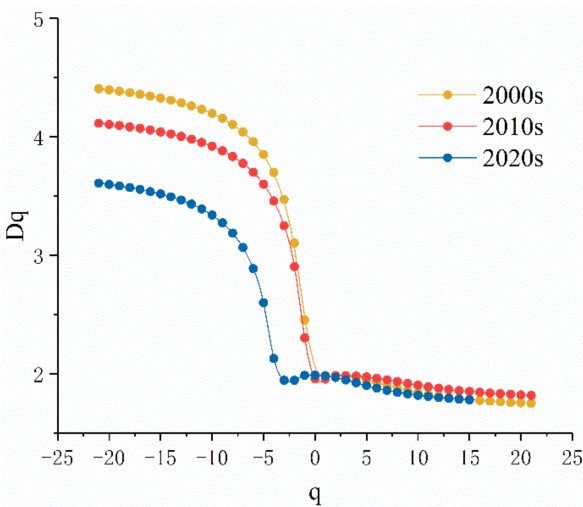

**Figure 7.** The relationship between generalized multifractal dimension $D_q$ and order moment $q$ in the 2000s, 2010s, and 2020s.

### 3.3. Multifractal Spectrum Analysis

The change in the river network was analyzed using the multifractal spectrum ($\alpha(q) \sim f(\alpha)$), which is a quantitative index used to describe the singularity, complexity, and self-similarity of the river network [36]. The multifractal spectrum and its characteristic parameters of the river network in the 2000s, 2010s, and the 2020s were calculated according to Equations (9)–(11), and the results are shown in Figure 8 and Table 1.

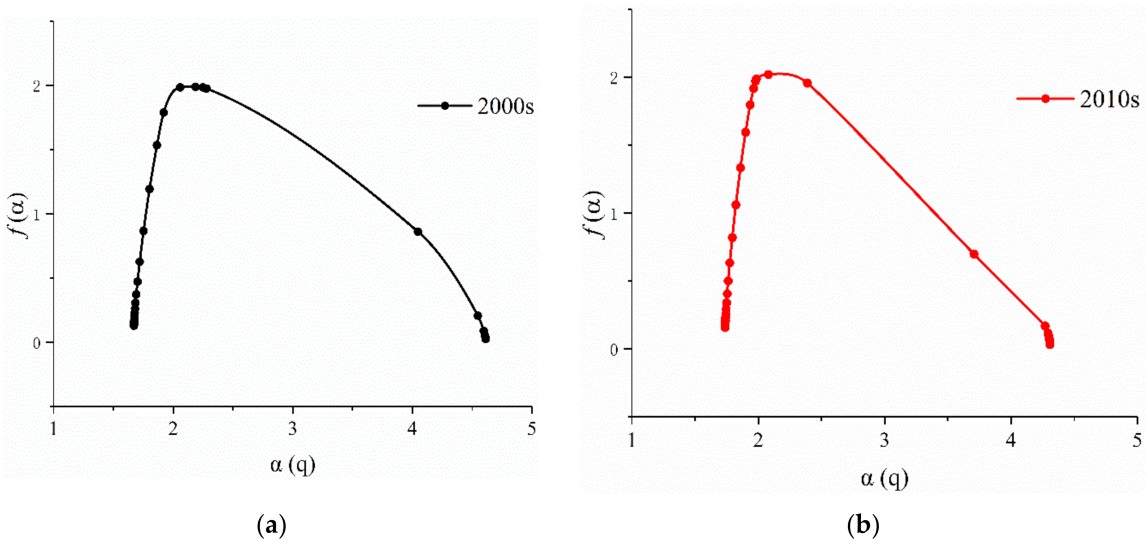

(**a**)          (**b**)

**Figure 8.** *Cont.*

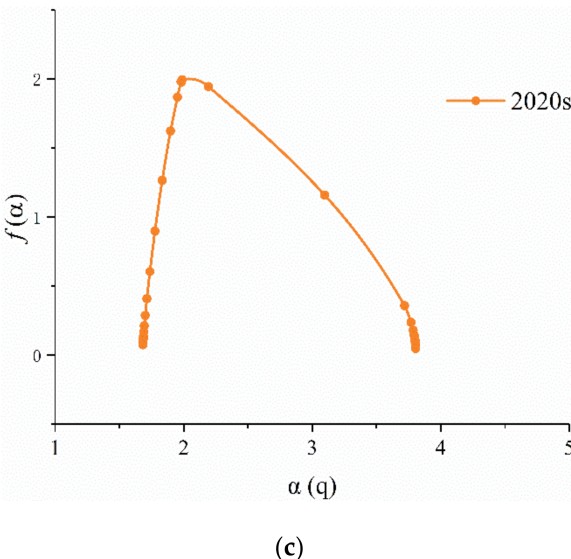

(**c**)

**Figure 8.** The multifractal spectrum of the Yellow River Basin in the (**a**) 2000s, (**b**) 2010s, and (**c**) 2020s.

**Table 1.** Multifractal characteristic parameters of the river network in the Yellow River Basin.

| Periods | $\alpha_{min}$ | $\alpha_{max}$ | $f(\alpha_{min})$ | $f(\alpha_{max})$ | $\Delta\alpha$ | $\Delta f$ |
|---------|---------|---------|---------|---------|---------|---------|
| 2000s | 1.6720 | 4.6134 | 0.1286 | 0.0262 | 2.9414 | 0.1024 |
| 2010s | 1.7376 | 4.3076 | 0.1568 | 0.0324 | 2.5700 | 0.1243 |
| 2020s | 1.6836 | 3.8057 | 0.0746 | 0.0474 | 2.1220 | 0.0272 |

The span $\Delta\alpha$ of the singular exponent can represent the unevenness of the spatial distribution of the river network in the basin. The larger the $\Delta\alpha$ is, the more uneven the distribution of the river network. As shown in Figure 8, the multifractal spectrum of the river network in the three periods of the Yellow River Basin are all asymmetric hook-shaped curves. The curves are all skewed to the left and the right side is dominant. The multifractal spectrum width $\Delta\alpha$ of the river network in the study area decreased from 2.9414 to 2.57 during 2000–2010, and then to 2.122 in the 2020s (Table 1). This indicates that the complexity of the river network in the basin decreased over time. In the multifractal spectrum, the side with a larger $\alpha$ value corresponds to the part of $q < 0$, which describes the advantage of the subset with a higher probability of characteristic information. That is, the region with a larger river network density. The side with a smaller $\alpha$ value corresponds to the part of $q > 0$, which describes the features of the region with a lower river network density. As shown in Figure 8, the fluvial structure of the study area was affected by, and highly dependent on, the river network density from 2000 to 2020. Meanwhile, the influence of density of the rivers in the network gradually decreased over time.

In addition, the $\Delta f$ in the multifractal spectrum describes how the river network structure is affected by abundant and sparse river densities. The $\Delta f$ of the river network in the Yellow River Basin increased from 0.1024 in the 2000s to 0.1243 in the 2010s, and then decreased to 0.0272 in the 2020s (Table 1). Temporally, the river network in the study area was most affected by high river densities, which is consistent with the conclusion obtained from the above analysis. The Yellow River Basin is mainly located in the plateau geomorphic area. In the past 20 years, the temperature has generally shown an upward trend, and the spatial difference in precipitation is significant [37]. The degree of desertification is above medium risk and intensifies from south to north. Additionally, the urbanization process has accelerated in recent years, which has led to a decrease in the studied river network (including river density and quantity) [38].

### 3.4. Correlation Analysis of Multifractal Indicators and the Urbanization Process

In order to analyze the multifractal characteristics of the spatiotemporal variation in the Yellow River Basin (YRB), it was divided into nine provinces: Qinghai Province (QH), Sichuan Province (SC), Gansu Province (GS), Ningxia Hui Autonomous Region (NX), Inner Mongolia Autonomous Region (NMG), Shaanxi Province (SA), Shanxi Province (SX), Henan Province (HN), and Shandong Province (SD) (Figure 1c). The Yellow River Basin is in northwestern China, including a part of the southern region. The dominant topography is a dissected plateau. In the past 20 years, more attention has been paid to high-quality development of the Yellow River Basin, which has led to its rapid economic development and urbanization. At the same time, existing research shows that, when examined in the short-term, the change in the characteristics of the river network is mainly affected by urbanization and the construction of water conservancy projects [38]. Accordingly, we consider the urbanization process as the main factor affecting the change in the river network and analyzed the impact of urbanization on the change of the river network in different areas of the Yellow River Basin.

The structure of a river network is affected by many factors, such as geology, topography, precipitation, temperature, and human activities. However, in the time range studied, the changes in the river network are mainly affected by human activities, especially urbanization [39]. To analyze the relationship between the urbanization process and the change in river network structure over time, we adopted the Gray correlation analysis. The Gray correlation analysis is a method to measure the degree of correlation between elements in time series, based on the degree of similarity or difference in development trends among the elements. It can be calculated using Equations (13) and (14):

$$\zeta_{0i} = \frac{\Delta(\min) + \rho\Delta(\max)}{\Delta_{0i}(k) + \rho\Delta(\max)}, \tag{13}$$

$$r_i = \frac{1}{n}\sum_{k=1}^{n}\zeta_i(k). \tag{14}$$

where $\zeta_{0i}$ is the correlation coefficient; $\Delta(\min)$ is the minimum value of the difference between the reference sequence and the mother sequence, $\Delta(\max)$ is the maximum difference value, $\rho$ is the resolution coefficient (0.5 in this study), $\Delta_{0i}(k)$ is the absolute value of the difference between the parent sequence and the reference sequence, $r_i$ is the value of Gray correlation, and $n$ is the number of correlation coefficients. In the calculation, the urbanization rate is the parent sequence, and the other parameters are the reference sequences.

In this study, the urbanization index used is the urbanization rate, and the urbanization rate of each province in the 2000s, 2010s, and 2020s was calculated. The data (urbanization rate of each province and the Yellow River Basin) used in the calculation were obtained from the China Statistical Yearbooks of the provinces in the Yellow River Basin (http://www.stats.gov.cn (accessed on 26 August 2021)). The Gray correlations between the change indexes of the river network and the urbanization process in the study areas were calculated. The indices included the river density $D_r$, multifractal spectrum width $\Delta\alpha$, multifractal spectrum difference $\Delta f$, capacity dimension $D_0$, and information dimension $D_1$. The calculation results are presented in Table 2.

**Table 2.** Gray correlation between the river network changes and the urbanization process of provinces in the Yellow River Basin.

| Urbanization Rate | Provinces | Parameters | | | | |
|---|---|---|---|---|---|---|
| | | $D_r$ | $D_0$ | $D_1$ | $\Delta\alpha$ | $\Delta f$ |
| Gray correlation | QH | 0.7900 | 0.7870 | 0.7935 | 0.7317 | 0.7843 |
| | GS | 0.7384 | 0.7380 | 0.7397 | 0.7355 | 0.7424 |
| | NX | 0.7417 | 0.7343 | 0.7391 | 0.7882 | 0.7303 |
| | SA | 0.7453 | 0.7417 | 0.7469 | 0.6977 | 0.7472 |
| | NMG | 0.7189 | 0.7239 | 0.7217 | 0.7193 | 0.7171 |
| | SX | 0.7698 | 0.7653 | 0.7615 | 0.7211 | 0.7634 |
| | HN | 0.7494 | 0.7481 | 0.7543 | 0.7055 | 0.7464 |
| | SD | 0.7783 | 0.7664 | 0.7701 | 0.7322 | 0.7657 |
| | SC | 0.8896 | 0.8563 | 0.8865 | 0.9326 | 0.8778 |
| | YRB | 0.7041 | 0.7159 | 0.7169 | 0.7967 | 0.7086 |
| | Average | 0.7626 | 0.7577 | 0.7630 | 0.7561 | 0.7583 |

As can be seen from Table 2, the values of the Gray correlation degree are all positive, and all are greater than the resolution coefficient ($\rho = 0.5$). The results indicate that there is a significant positive correlation between the river network change and the urbanization process. That is, the higher rate of urbanization of the region, the greater the impact on the river network. According to the Statistical Yearbook of China, the urbanization rate of the Sichuan Province (SC) changed the most among the nine provinces in the Yellow River Basin during the period 2000–2020, which indicates that its river network structure was also affected the most by the urbanization process. In general, the average Gray correlation of changes in the river characteristic parameters is as follows: $D_r > D_1 > \Delta f > D_0 > \Delta\alpha$, indicating that the density of the river network is greatly affected by the urbanization process. These conclusions also confirm that in the short-term, the urbanization process has a greater impact on the structure of the river network.

## 4. Discussion

In this study, multifractal characteristics of the study area in the 2000s, 2010s, and the 2020s were calculated, which revealed the spatiotemporal variation characteristics of the river network. In order to better study the spatiotemporal change characteristics of the river network in the Yellow River Basin, the multifractal characteristics and the river network density of the studied provinces were calculated in the 2000s, 2010s, and 2020s. Additionally, we analyzed the change rate of parameters during the period 2000–2010 and 2010–2020 according to the calculation results. Results are shown in Figure 9.

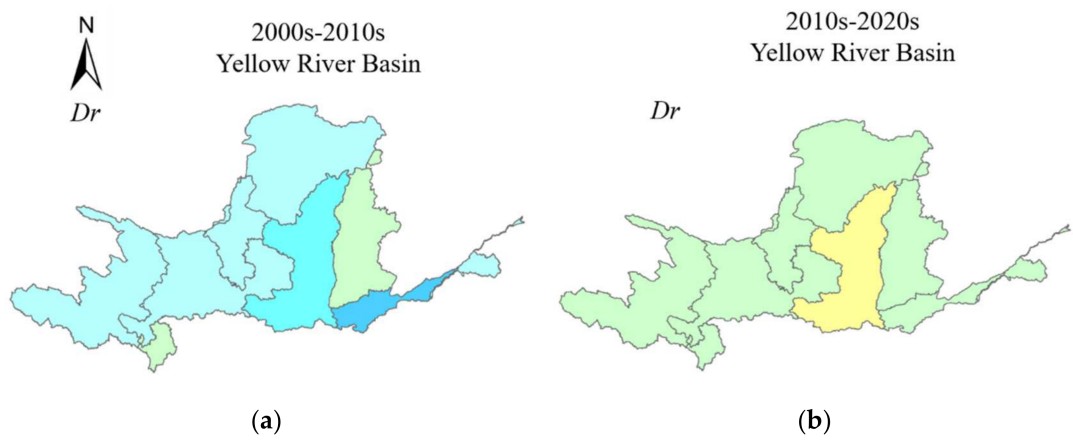

(**a**) (**b**)

**Figure 9.** *Cont.*

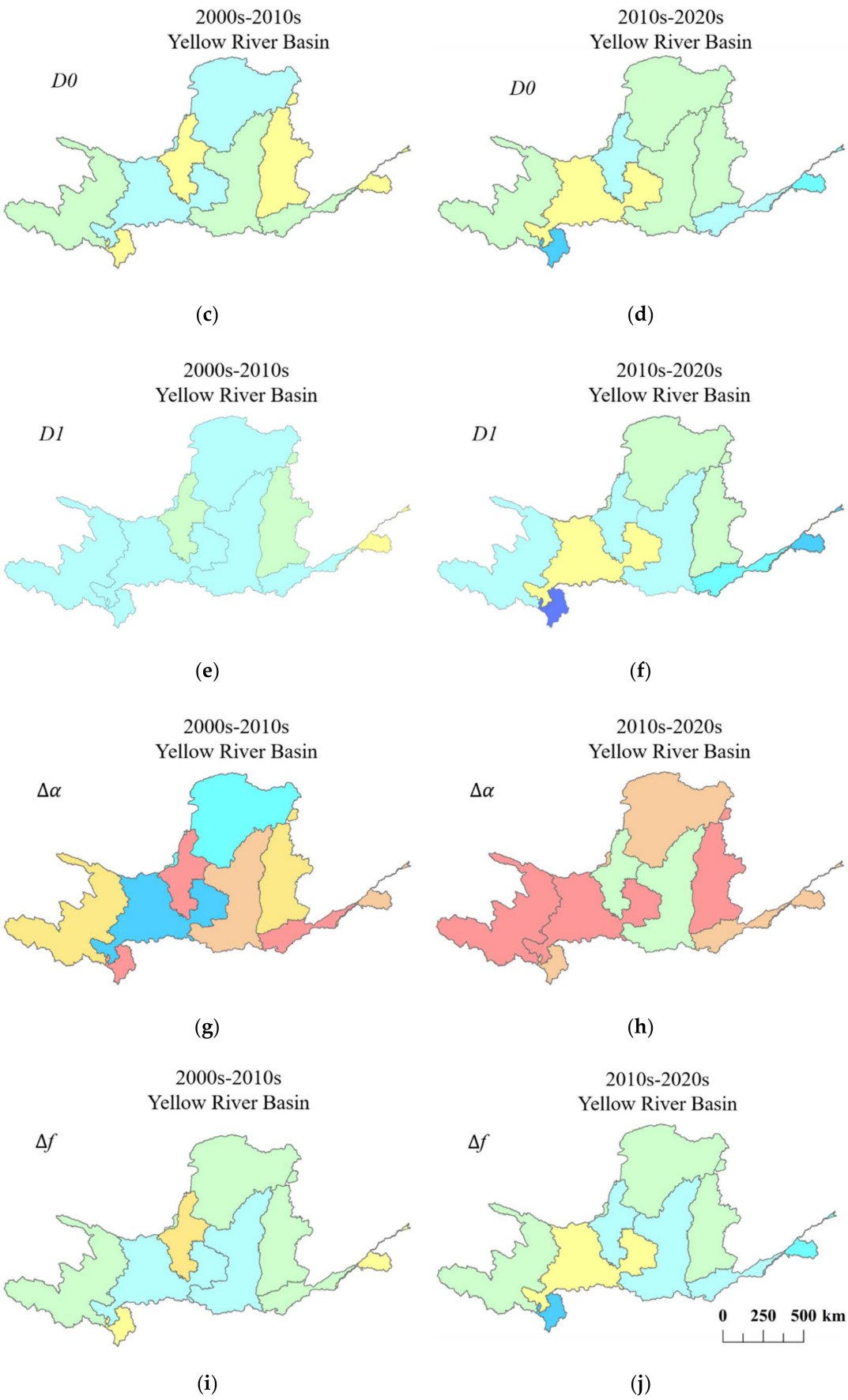

**Figure 9.** *Cont.*

## Legend

**Rate of change ( % )**

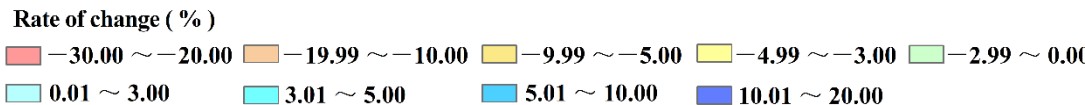

| | |
|---|---|
| $\square$ $-30.00 \sim -20.00$ | $\square$ $-19.99 \sim -10.00$ $\square$ $-9.99 \sim -5.00$ $\square$ $-4.99 \sim -3.00$ $\square$ $-2.99 \sim 0.00$ |
| $\square$ $0.01 \sim 3.00$ | $\square$ $3.01 \sim 5.00$ $\square$ $5.01 \sim 10.00$ $\square$ $10.01 \sim 20.00$ |

**Figure 9.** The variation of river characteristic parameters in provinces of the Yellow River Basin during the period 2000–2010 and 2010–2020. The change rate of river density in the period (**a**) 2000–2010 and (**b**) 2010–2020; the change rate of capacity dimension $D_0$ in the period (**c**) 2000–2010 and (**d**) 2010–2020; the change rate of information dimension $D_1$ in the period (**e**) 2000–2010 and (**f**) 2010–2020; the change rate of multifractal spectrum width $\Delta \alpha$ in the period (**g**) 2000–2010 and (**h**) 2010–2020; the change rate of multifractal spectrum difference $\Delta f$ in the period (**i**) 2000–2010 and (**j**) 2010–2020.

According to the commonly used classification method of urbanization stages [40] and urbanization rate of the Yellow River Basin, we divide the study area into two urbanization stages through calculation: Those periods with an urbanization rate of less than 45% are classified as the slow urbanization stage, and those with an urbanization rate of 45% to 65% are classified as the rapid urbanization stage. Therefore, according to the urbanization rate of the Yellow River Basin, it can be divided into a slow urbanization stage (2000–2010) and a rapid urbanization stage (2010–2020). It can be seen from Figure 9 that in the past 20 years, the spatial and temporal variations of the river network parameters are significantly different at different stages of urbanization. Overall, the values of the characteristic parameters decreased in the studied provinces of the Yellow River Basin, but the degree of decrease is not identical in different stages of urbanization. The amount of reduction of each parameter in the slow urbanization stage was less than that in the rapid urbanization stage, and the values of some parameters increased during the slow urbanization stage.

In general, the river network density increased during the period of 2000–2010. However, there were some areas where the river density decreased (such as the Shanxi and Sichuan Provinces during 2010–2020). Compared with other areas, river density in the Shaanxi Province decreased significantly. In this study, we used the fixed-size box counting algorithm (FSA) to calculate the multifractal features of the river network, so the capacity dimension $D_0$ is the box dimension. The box dimension of the river network in the Yellow River Basin decreased during the period 2000–2020, and the rate of decrease in the slow urbanization period was less than that in the rapid urbanization period. Compared to the information dimension $D_1$, the capacity dimension $D_0$ was reduced to a greater degree. The variation in the multifractal spectrum width $\Delta \alpha$ of the river network in the Yellow River Basin varies greatly from 2000 to 2020, but the rate of change generally decreased. The minimum rate of change is less than 1% (Ningxia Hui Autonomous Region, in the period 2010–2020), and the maximum change rate is about 30% (Qinghai Province, in the period 2010–2020). This shows that the complexity of the river network in each province was reduced. The decreasing rate of the multifractal spectrum width $\Delta \alpha$ of the river network in each province is higher in the rapid urbanization stage than in the slow urbanization stage. Its change is most obvious compared to other fluvial characteristic parameters, such as river density. The result of the multifractal spectrum width shows that the multifractal analysis is more sensitive to changes in the river network. The difference $\Delta f$ in the multifractal spectrum of the river network in the study area was greater than 0. The difference $\Delta f$ generally decreases over time, but changes to different degrees in different regions. The degree of change in the rapid urbanization stage is higher than that in the slow urbanization stage. Based on the change in the $\Delta f$ in the multifractal spectrum, it can be concluded that the river network structure of the provinces in the Yellow River Basin is mainly affected by densely distributed rivers. Although the influence of dense rivers decreased overall, there were some areas where the influence of dense rivers increased, such as the Sichuan Province and the Shandong Province.

During the urbanization process from 2000 to 2020, the spatial and temporal differences in the river network changes in the Yellow River Basin were clear. The main reason

for these changes and the temporal and spatial differences are that they are affected by the urbanization process. Although natural factors such as temperature and precipitation also have certain impacts on the change of the river network structure, the impacts are small, owing to the small span of research time. In recent years, the economic development of central and western China has increased. The Yellow River Basin is mainly located in northwest China, and its economic center has a tendency to expand to the northwest. Therefore, the degree of urbanization of the Yellow River Basin has significantly increased over the past 20 years [41]. The differences in the river network of the Yellow River Basin as a result of urbanization are large but show a decreasing trend. The difference in degrees of urbanization is also an important factor that causes temporal and spatial differences in river network changes. Furthermore, the degree of river reconstruction is different at different stages of the urbanization process (periods of rapid urbanization and slow urbanization). In the rapid urbanization stage, human activities are more frequent, and the construction of reservoirs and other water conservancy projects (such as Qingtongxia water conservancy Projects) have a greater impact on the river structure [42,43]. In addition, in order to expand the area of urban utilization, measures such as canalisation, straightening of meanders, and filling and excavation of the river have been carried out. In order to deal with disasters such as urban waterlogging, the surface runoff has been managed and sponge cities have been built. In the Yellow River Basin, famous hydropower stations such as Xiaolangdi and Sanmenxia were built in order to use the rivers more fully [42]. All these measures of urbanization have directly or indirectly affected the structure of the river network. This is consistent with the conclusions of this study.

## 5. Conclusions

In this study, we computed and analyzed the characteristics of river network changes in the Yellow River Basin over the past 20 years using multifractal analysis. Based on the vector data of the studied river network, we calculated the generalized multifractal dimension ($D_q$) and multifractal spectrum ($f(\alpha)$) of the study area using multifractals and analyzed the results. Moreover, we analyzed the multifractal characteristics of nine provinces in the Yellow River Basin and discussed the urbanization driving changes in the river network structure. The conclusions obtained are as follows:

1. During the period of 2000–2020, the river network of the Yellow River Basin has clear multifractal properties. It was found that the river network structure of the Yellow River Basin is greatly affected by areas of higher river density. The river network structure (the number and density of the rivers in the network, etc.) has shown a decreasing trend over the past 20 years, and the degree of the impact of dense rivers has also decreased.

2. The changes in river networks were significantly affected by urbanization. Changes in river network structure were significantly correlated with the urbanization process. The average Gray correlation values between the changes in river networks and urbanization were greater than 0.7, which was greater than the resolution coefficient of the Gray correlation analysis (0.5). Their order was $D_r > D_1 > \Delta f > D_0 > \Delta \alpha$. This result indicates that the greater the urbanization rate, the greater the impact on the river network structure.

3. To better study the spatiotemporal characteristics of river network changes in the Yellow River Basin in the context of urbanization, we calculated the fluvial characteristic parameters of provinces in the study area during periods of slow urbanization (2000–2010) and rapid urbanization (2010–2020). Moreover, we analyzed the degree of variation and temporal and spatial differences in these parameters. The results show that the changes in the river network structure are more affected by urbanization during the rapid urbanization stage. The multifractal spectrum width $\Delta \alpha$ is more sensitive to changes in the river network structure.

In general, the structure of a river network is highly complex and self-similar, and the development of river networks is determined by many factors. Therefore, it is difficult to

construct an accurate model to describe and predict its features. Multifractal theory can be used as an efficient means for evaluating and predicting changes in river networks. It is of great theoretical significance to quantitatively describe and understand a river network structure and its changes.

**Author Contributions:** Conceptualization, Z.Q. and J.W.; methodology, Z.Q.; software, Y.S., and J.Y.; resources, Z.Q.; data curation, J.W.; writing—original draft preparation, Z.Q.; writing—review and editing, Z.Q., J.W., Y.S., and J.Y.; funding acquisition, J.W. All authors have read and agreed to the published version of the manuscript.

**Funding:** This research was funded by the Key Scientific and Technological Project of Henan Province, China (grant number 212102210377).

**Institutional Review Board Statement:** Not applicable.

**Informed Consent Statement:** Not applicable.

**Data Availability Statement:** The data presented in this study are available on request from the corresponding author.

**Acknowledgments:** We would like to thank all editors and reviewers for their insightful comments, which helped us improve the quality of this paper.

**Conflicts of Interest:** The authors declare no conflict of interest.

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
