# Peer review of "Multifractal Analysis of River Networks under the Background of Urbanization in the Yellow River Basin, China"

_water, doi:10.3390/w13172347_

Round 1

Reviewer 1 Report

1)introduction: The scientific question is not clear. “As one of the indicators of the complexity of a river network structure, the multifractal spectrum has not only been well applied in the study of river network structure and changes, but has also been used to improve the predictability of runoff and rainfall models.” &” However, in recent decades, the structure or development of many river networks has been greatly affected by large-scale urban expansion, which has had a strong anthropogenic impact on the surface of the earth [20]. Under the influence of urbanization, more than half of the world’s river network structures have undergone changes to varying degrees.” What is the value of this article? Only “but there are few temporal and spatial analysis studied for large river basins”?

2)Will there be any difference between the method applied in large watershed and small watershed?

3) Abstract: “We further applied multifractal analysis to study the river network structure changes”. What does further mean?

4) The focus of this paper should be “Multifractal Analysis of River Networks under the Background of Urbanization”, That is, the relationship between urbanization and Multifractal indicators. So, some content should be put in the results rather than in the discussion.

5) “According to the commonly used classification method of urbanization stages [40], those periods with an urbanization rate of less than 45% are classified as having slow urbanization, and those with an urbanization rate of 45% to 65% are classified as having rapid urbanization. Therefore, the urbanization rates of the Yellow River Basin can be divided into a slow urbanization stage (2000-2010) and a rapid urbanization stage (20102020).” Maybe the classification result of stage is right, but the basis is not sufficient. Why not use the calculated urbanization index? What is the urbanization index used in this paper?

6) What is the ecological or practical meaning (significance) of the multifractal index? The conclusions “the higher the rate of urbanization, the greater the impact on the river network structure” based on statistics do not have much significance.

7) “During the period 2000-2020, the urbanization rate of Sichuan Province (SC) changed the most among the nine provinces in the Yellow River Basin.” Why? Is it because of its small area or because it is mainly a wetland? How to ensure the consistency and accuracy of river network extraction?

Author Response

Dear Reviewer,

Thank you for your comments concerning our manuscript entitled “Multifractal Analysis of River Networks under the Background of Urbanization in the Yellow River Basin, China” (ID: water-1322627). We appreciate the time and effort that you dedicated to providing feedback on our manuscript and are grateful for the insightful comments. We have studied comments carefully and have made the correction which we hope meet with approval.

The main corrections are as follows:

  1. In order to better express the value of the article, we have corrected the content of this part, the revised content can be found at the end of the introduction.
  2. The content of “4.1 Correlation Analysis of River Network Change and the Urbanization Process” has been put in the “Results”, and We have adjusted the content of the “Discussion”.
  3. The division of different urbanization stages was based on the existing research, and the division results are obtained according to the calculation formula and the urbanization rate of the study area. The classification method in this paper is a common method, and this method has been verified and widely used. The different urbanization rate in different regions leads to the different results of the division stage. In this paper, the urbanization index used in this paper is the “urbanization rate”. The data of urbanization rate is obtained from the China Statistical Yearbook (mentioned in 3.4 of the paper), which is the public data on the urbanization rate of the Chinese government and uses this index to indicate the degree of urbanization. In order to make this clear, this statement has been corrected.
  4. In order to ensure the consistency and accuracy of river network extraction, digital data of the studied river networks during the 2000s and the 2010s were obtained by digitizing 1: 50,000 digital line graphics, for the 2020s, data were derived from OpenStreetMap, and the river network of the study area was corrected by adding and deleting the river networks in three periods based on Google Earth remote sensing images.

     5. The specific corrections can be found in the attachment.

Once again, thank you very much for your warm work.

Best regards.

Reviewer 2 Report

The authors present a description of the use of multifractal analysis as a tool to analyze the changes in a river basin network, utilizing the Yellow River Basin as a case study. The paper is very well written and the concept is clearly described. The references are current and appropriate, and figures and tables necessary for the paper. 

One observation regarding the discussion is that it is more results oriented: some description of what is meant by urbanisation, including reference for example to land use plans, would be helpful for readers unfamiliar with the Basin. This reviewer assumed that urbanisation would include canalisation, straightening of meanders, and perhaps alteration of the conveyance systems and stormwater/runoff management practices. The authors also mention water conservancy projects, but do not elaborate. Clearly, the data show significant changes in the river system during the last 20 years, which was the intent of the analysis.

Author Response

Dear Reviewer,

Thank you for your comments concerning our manuscript entitled “Multifractal Analysis of River Networks under the Background of Urbanization in the Yellow River Basin, China” (ID: water-1322627). We appreciate the time and effort and are grateful for the insightful comments. Those comments are valuable and very helpful for revising and improving our paper. We have studied comments carefully and have made the correction which we hope meet with approval.

In order to express the content of urbanization process more clearly and in detail, we added some contents about urbanization in the section of discussion. We listed the large water conservancy projects built in recent years and the management of surface runoff, etc.

Once again, thank you very much for your warm work.

Best regards.

Round 2

Reviewer 1 Report

Point 7: The part of Sichuan in Yellow River basin accounts for only a small part of the whole province. Using the urbanization rate of Sichuan Province appropriate?

Author Response

Dear Reviewer,

Thank you for your comments concerning our manuscript entitled “Multifractal Analysis of River Networks under the Background of Urbanization in the Yellow River Basin, China” (ID: water-1322627). We have studied comments carefully and have made the explanation which we hope meet with approval.

The urbanization rate of Sichuan Province is a statistical value based on the whole province. Therefore, it can represent the urbanization rate of Sichuan province in the Yellow River Basin. More accurate and detailed statistics and statistical results are also what we are pursuing, but as a result of published by the provinces, municipal statistics are discontinuous or missing (unpublished), which makes us will not be able to use the city-level urbanization rate for analysis (e.g., Mianyang City and Ya'an City), so we use the urbanization rate of Sichuan Province as the urbanization rate of Sichuan Province in the Yellow River Basin, and it is appropriate to use the urbanization of Sichuan Province to analysis in this paper.

Once again, thank you very much for your warm work.

Best regards.

This manuscript is a resubmission of an earlier submission. The following is a list of the peer review reports and author responses from that submission.